# Automated Approach for Computer Vision-Based Vehicle Movement Classification at Traffic Intersections

Udita Jana [1],*, Jyoti Prakash Das Karmakar [1], Pranamesh Chakraborty [1], Tingting Huang [2] and Anuj Sharma [3]

[1]  Department of Civil Engineering, Indian Institute of Technology Kanpur, Kanpur 208016, India
[2]  ETALYC Inc., Ames, IA 50010, USA
[3]  Department of Civil, Construction, and Environmental Engineering, Iowa State University, Ames, IA 50011, USA
*  Correspondence: uditajana20@iitk.ac.in

**Abstract:** Movement-specific vehicle classification and counting at traffic intersections is a crucial component of various traffic management activities. In this context, with recent advancements in computer-vision-based techniques, cameras have emerged as a reliable data source for extracting vehicular trajectories from traffic scenes. However, classifying these trajectories by movement type is quite challenging, as characteristics of motion trajectories obtained this way vary depending on camera calibrations. Although some existing methods have addressed such classification tasks with decent accuracies, the performance of these methods significantly relied on the manual specification of several regions of interest. In this study, we proposed an automated classification method for movement-specific classification (such as right-turn, left-turn and through movements) of vision-based vehicle trajectories. Our classification framework identifies different movement patterns observed in a traffic scene using an unsupervised hierarchical clustering technique. Thereafter, a similarity-based assignment strategy is adopted to assign incoming vehicle trajectories to identified movement groups. A new similarity measure was designed to overcome the inherent shortcomings of vision-based trajectories. Experimental results demonstrated the effectiveness of the proposed classification approach and its ability to adapt to different traffic scenarios without any manual intervention.

**Keywords:** movement classification; trajectory analysis; hierarchical clustering

## 1. Introduction

Traffic safety is one of the major transportation concerns in the world and in the USA. Traffic intersections are one of the critical hotspots for crashes and fatalities. In 2019, out of the 36,096 traffic fatalities, approximately 28% were related to traffic intersections and junctions alone [1]. This can be attributed due to the complex interactions between vehicles and pedestrians that lead to conflict points and potential accident hotspots. Moreover, traffic intersections are major bottlenecks of the traffic system and a primary source of traffic delays, too. Therefore, traffic intersection monitoring and management are crucial steps for improving traffic safety and mobility.

One of the first steps of smart traffic intersection monitoring is the efficient collection of traffic data, which includes the total vehicle counts in the intersection and the classification of turning movements and through movement for the different intersection approaches. Turning and through movement counts are essential for traffic analysis activities such as signal-timing optimization and congestion management. These movement-specific vehicle counts often guide the decision-making process followed by transportation agencies for tackling complex traffic management problems. As a result, there is a significant demand for such movement count data.

Inductive loop detectors, radar sensors, and Bluetooth devices are the common sources that provide traffic count data in general [2–4]. Inductive loop detectors can provide

reliable traffic movement counts. However, they are intrusive, and their installation or repair work is time consuming and can require stopping traffic movements, too. On the other hand, road-side radars are popularly used non-intrusive sensors, as they can provide accurate information about vehicle trajectories. Another such alternative is traffic surveillance cameras. These devices are commonly installed for monitoring purposes at traffic intersections. Recently, computer-vision-based techniques aided by deep learning methods have shown promising potential in object detection and classification tasks [5,6]. Therefore, we can also produce similar vehicle trajectory information from existing traffic surveillance cameras. Video data obtained from surveillance cameras have been also utilized for real- time traffic forecasting or safety analysis purpose [7–10]. Compared to radars and loop detectors, traffic cameras can provide a wide field of view at a much lower hardware cost. The movement counts generated from radars and loop detectors assume lane-disciplined driving behavior, which is not always prevalent in complex traffic scenarios; this is a major disadvantage. In contrast, computer vision-based approaches are not limited by such strict lane-adherence behavior; therefore, these methods can be leveraged for obtaining the traffic movement counts even in challenging driving scenes.

Trajectory data extracted from vision-based systems record vehicle locations in terms of image coordinates owing to which the appearance of vision-based motion trajectories can change depending on the extrinsic calibration of the camera. Several studies on vehicle movement prediction [11] and traffic incident detection have utilized similar vision-based trajectories. Mainly supervised approaches [12] and few semi-supervised [13] or unsupervised approaches were undertaken in these studies. Undoubtedly, supervised methods can provide better accuracies for such classification tasks. However, opting for these approaches is not a feasible alternative in our case, as it will involve generating a large number of data labels manually. Similar approaches have also been deployed in video-based activity recognition tasks as well [14–18].

Regarding the movement classification task, the AI City Challenge [19] focused on generating classified movement counts from traffic surveillance videos. While different strategies were adopted for achieving efficient and effective classification and they provided decent accuracies, there are some major shortcomings. These methods were implemented on a particular region of interest (ROI), which was specified manually. Regardless of the classification scheme, these methods utilized manually identified features (i.e., specific entry–exit region and/or trajectory pattern for a particular movement). Most often, these were rule-based classification schemes for a specific traffic scenario. As a result, such classification schemes cannot readily adapt to different traffic scenes without any manual intervention. The manual effort required at individual traffic scenes, although negligible, is an essential component of these classification schemes, owing to which they were not transferable. Therefore, these approaches have limited application in large-scale city-wide deployment.

To address the shortcomings mentioned above, this study proposes an automated method, particularly for computer-vision-based movement classification, using an unsupervised learning approach. In this method, the relevant features guiding the classification process are automatically identified in each traffic scene. The proposed approach follows an unsupervised learning scheme and therefore is adaptive to different traffic scenarios. It consists of four distinct steps, the first being detection of the stopping location for vehicles in intersection approaches followed by clustering the vehicle trajectories into movement clusters using unsupervised clustering. Then, the clusters are used to extract modeling trajectories for each movement, which can be compared to the incoming trajectories to classify them accordingly. The major advantage of the proposed approach is that it is an automated method, which can be applied to any traffic intersection approach to perform movement classification, thereby being easily scalable to large city-wide applications with minimal manual intervention.

The following section gives an overview of previous research on video-based vehicle trajectory classification. The third section specifies the details of our proposed classification

algorithm. Section 4 contains the description of the dataset used in this study followed by experimental results and discussions in Sections 5 and 6. The final section provides a summary of the paper and briefly outlines future works.

## 2. Related Work

The first step toward vision-based movement classification is trajectory formation using object detection and tracking algorithms. In the deep learning era, object detection and tracking made tremendous progress both in terms of accuracy and efficiency. CNN algorithms such as Faster R-CNN [20] and Mask R-CNN [21] have achieved state-of-the-art accuracies in object detection tasks. However, detection modules such as YOLO [22], SSD [23], and RetinaNet [24] are more efficient and thereby preferred for robust real-time object detection. Various research works are going on to have a better trade-off between accuracy and efficiency. Tracking after detection is a popular strategy adopted for multi-object tracking (MOT). Outputs obtained from object detection algorithms across video frames are used in the tracking phase, which generates the trajectories of the objects. These tracking algorithms provide spatial as well as temporal information of vehicular movement within the video frames. SORT [25] and DeepSORT [26] are some of the popular alternatives that provide decent accuracies in MOT problems.

Existing studies that addressed vehicle movement classification tasks from video data can be broadly classified into three categories: Line-crossing, Zone-traversal, and Trajectory similarity-based classification.

- The first one follows a line-crossing based approach [27,28] where a virtual entry and exit line pair is considered for identifying each movement of interest (MOI) inside a prespecified region of interest (ROI). This method is not very effective for classifying incomplete trajectories resulting from identity switches or occlusions.
- Studies belonging to the second category [29–31] are similar to the first one, but instead of line pairs, several virtual zones (either entrance or both entry–exit zone pairs) are manually selected depending on traffic scenarios. Although these studies produced better accuracies compared to line-crossing-based methods, they were not very efficient for unstructured driving environments.
- The third group of studies [32,33] classified vehicle movements following a similarity-based assignment. Representative trajectories for different MOIs were manually chosen for this purpose. These similarity-based methods were found to be much more efficient for classifying incomplete vehicle tracks. However, they are not scalable when it comes to adapting to various complex traffic scenarios.

Unsupervised methods are generally preferred for increasing scene adaptability. In this regard, hierarchical clustering-based techniques have been used for classifying vehicular trajectories [34,35]. The Hausdorff distance between trajectory pairs has been predominantly used as a similarity measure for these clustering techniques [36]. The traditional Hausdorff distance has certain drawbacks for comparing trajectories of unequal length, as it is sensitive to noise and does not account for the direction of ordered data pairs. Various improvements [37,38] have been proposed to remove noise sensitivity and incorporate directional features to the traditional Hausdorff distance. Therefore, unsupervised techniques can be a good alternative toward automated movement classifications.

## 3. Methodology

The first step toward video-based vehicular movement classification is to extract vehicle trajectories, which involves object detection and tracking processes. A significant amount of past research has addressed the vision-based detection-tracking problem, which is already highlighted in the previous section. Therefore, our study mainly focused on the classification of vehicular trajectories to the corresponding movement types (i.e., through, right turn, and left turn). For obtaining the vehicle trajectories, state-of-the-art detection and tracking algorithms have been utilized, the details of which are provided in the data description section.

As discussed in the previous section, earlier attempts in this vision-based movement classification task have achieved good accuracies. However, these methods significantly depend on manual annotations for specifying the entry and exit regions of each movement type. Since this study gears toward obtaining automated movement-specific vehicle counts at traffic intersections, we have focused on unsupervised methods for addressing such classification tasks.

### 3.1. Proposed Classification Algorithm

The proposed movement classification algorithm requires vehicle trajectories. In this study, we have extracted visual trajectories using a detection-by-tracking framework, which involves using a vehicle detector and a tracker at a subsequent stage therefore formulating the trajectory. Details regarding this framework are mentioned in Section 4. The proposed classification algorithm comprises four stages. At first, the 'stopbar' is located near each intersection approach. In the next stage, vehicle trajectories are clustered according to their movement type, which is then followed by modeling trajectory selection. In the final stage, each incoming vehicle's trajectory is compared to modeling trajectories by a pre-defined similarity measure and is classified accordingly.

#### 3.1.1. Stopbar Identification

This is the first stage of the proposed algorithm. Depending on the coverage of the vision-based detection system, vehicles may be detected and tracked from a significantly earlier stage even before their arrival near turning locations. Thus, the detection-tracking algorithms generate a significant amount of additional information for each incoming vehicle trajectory. Such additional information is irrelevant for the classification purpose.

Moreover, the inclusion of such additional information adversely affects the performance of the clustering algorithms and makes the clustering stage more time consuming and cumbersome. To avoid these complications, the concept of the stopbar, which was introduced by Santiago-Chaparro et al. [39] for radar-based vehicle trajectory analysis, has been adapted and applied in this study for vision-based stopbar identification.

'Stopbar' as defined in this stage does not necessarily align with the lane markings (i.e., stop lines) or stop signs placed at the intersection legs. Rather, it provides a general understanding of where vehicles are diverging at each incoming approach, which is useful for identifying relevant patterns of different movement types.

For placing the stopbar, first, stopped locations were extracted from tracking outputs obtained for a particular site. In this study, vehicle positions with no displacement in two consecutive video frames were chosen as the stopped location. A horizontal line passing through the 50th percentile value of y-coordinates of these stopped locations was then marked as the stopbar.

Since the spread of the stopped location is not only limited around the actual stop-line marks (refer to Figure 1b), choosing a stopbar at the lower quartile value will retain a lot of additional information. Alternatively, opting for the upper quartile or a higher quantile may result in discarding relevant movement patterns that will aid the classification process. Considering the above aspects, the 50th percentile value was used to filter the relevant movement patterns.

A visual representation of how stopbars are located from raw trajectory data is shown in Figure 1. Figure 1a shows all trajectory points obtained from detection and tracking at a sample intersection approach, and Figure 1b shows the "stopped locations" for all trajectories as red points and the corresponding "stopbar" line. The vehicle trajectory points shown in the figure are with respect to video frame coordinates.

After locating the stopbar (denoted by $Y = Y_{sl}$), only the relevant portions of vehicle trajectories present after the stopbar, i.e., in the $Y \geq Y_{sl}$ region, is extracted and stored as a valid trajectory set $[T_{vs}]$. This valid trajectory set $[T_{vs}]$ is further used in the clustering phase. Since the cameras capturing the traffic movements are facing toward the intersection approach, the $Y \geq Y_{sl}$ part, i.e., the region after the stopbar in Figure 1b ($Y = 0$ marks the

top edge of the video frame), is chosen in this case. The next stage involves identifying the movement clusters from the trajectories extracted after the stopbar.

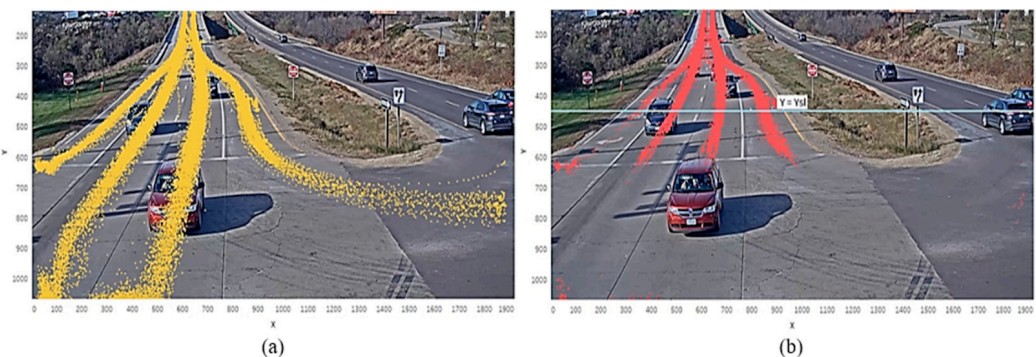

**Figure 1.** Stopbar identification process from vehicle trajectories at a sample intersection approach: (**a**) trajectory points, (**b**) stopped locations (red points) and the stopbar line.

### 3.1.2. Identifying Movement Clusters

The choice of clustering algorithm and choice of similarity measure are the two major factors that influence the classification process while using unsupervised methods. Clustering algorithms such as hierarchical clustering and Density-Based Spatial Clustering of Applications with Noise (DBSCAN) have been mainly used in previous studies for trajectory analysis purposes [40–42]. The DBSCAN algorithm, which has shown promising potential in outlier or anomalous trajectory detection, requires the setting of two hyperparameters (epsilon and min points). Selecting appropriate values of these hyperparameters provides site-specific solutions and will result in low-scene adaptability. On the other hand, hierarchical clustering only requires the number of clusters as input, which makes it more suitable for the movement classification task of our interest.

(a) Clustering algorithm—Owing to the reasons stated above, we have used an agglomerative clustering algorithm for identifying the movement clusters.

(b) Similarity measure—After the selection of the clustering algorithm, the next critical aspect of trajectory clustering is the choice of the similarity measure. In this study, three factors have been considered while designing the similarity measure.

▪ Distance similarity—At traffic intersections, vehicles moving in a particular direction generally traverse through a fixed region. Therefore, one of the important aspects to consider for movement classification is to check the spatial proximity between two trajectories. The Hausdorff distance has been widely used in the literature [36,37] for comparing the spatial proximity between trajectories. The first step for computing the traditional Hausdorff distance involves the calculation of the directed Hausdorff distance between a trajectory pair. This directed Hausdorff distance is defined as

$$d_H(P,Q) = \max_{p \in P} \left\{ \min_{q \in Q} d(p,q) \right\} \qquad (1)$$

Here, 'p' and 'q' stand for data points (vehicle positions in this case) of trajectory 'P' and 'Q', respectively. $d(p,q)$ is the Euclidean distance between p and q. To illustrate the directional Hausdorff distance, Figure 2 shows two sample trajectories, P and Q. The data points (vehicle positions) in these two trajectories are denoted by 'p' and 'q', respectively. Therefore, d(p,q) denotes the Euclidean distance between any two points, p and q. To estimate $d_H(P,Q)$, first, for each point p in trajectory P, the minimum Euclidean distance is computed for every point q of trajectory Q. This is denoted as $d(p_i,Q) = \min_{q \in Q} d(p,q)$. Then, the maximum of $d(p_i,Q)$ over all data points p of trajectory P gives the directional Hausdorff distance $d_H(P,Q)$, given by Equation (1) and also shown

in Figure 2. Similarly, $d_H(Q, P)$ can be calculated by first determining $d(q_i, P)$ and then finding its maximum over all data points q of trajectory Q, as shown in Figure 2, e.g., for the two trajectories 'P' and 'Q', as shown in Figure 2, $d(p_2, q_4)$ is the directed Hausdorff distance from trajectory 'P' to 'Q'. In contrast, the Euclidean distance between point $q_3$ and $p_2$, i.e., $d(q_3, p_2)$ marks the directed Hausdorff distance from trajectory 'Q' to 'P'.

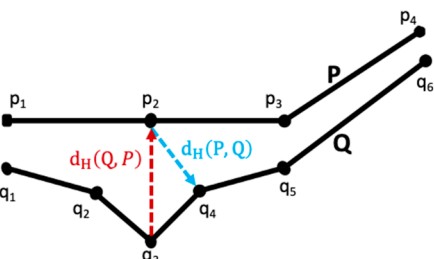

**Figure 2.** Directed Hausdorff distance between trajectories.

Two different vehicle trajectories can have different numbers of data points. In addition, these data points are not always uniformly spaced. Therefore, the values $d_H(P, Q)$ and $d_H(Q, P)$ are not always equal. The traditional Hausdorff distance returns the maximum between two directed Hausdorff distances computed between a trajectory pair. Since our intended classification task deals with trajectories of dissimilar lengths and incomplete trajectories, choosing the maximum value will not be suitable, as it will lead to very high distance separation values between two trajectories of dissimilar length even though they belong to the same movement class. For this reason, the reverse condition (i.e., minimum value) has been adopted for this study. Therefore, the distance similarity between two trajectories 'P' and 'Q', denoted by $D_S(P, Q)$, is defined as,

$$D_S(P, Q) = \min(d_H(P, Q), d_H(Q, P)) \tag{2}$$

where $d_H(P, Q)$ is the directed Hausdorff distance from trajectory 'P' to 'Q' as defined in Equation (1).

■ Angular similarity: For traffic intersections, the length of obtained vehicle trajectories varies significantly depending on their movement direction (right, left, through) and camera coverage. Along with this, the presence of broken or incomplete trajectories also produces vehicle trajectories of unequal length even within a specific movement group. Therefore, considering the distance similarity alone is not sufficient for such movement classification purposes. For estimating the directional difference between trajectories where incomplete trajectories of varying lengths might be present across different movement clusters, the following considerations were made.

In this formulation, $\overrightarrow{V_P}$ denotes the vector pointing from starting position to the end position of vehicle trajectory 'P'.

In addition, $\overrightarrow{V_{Q-P}}$ is the vector pointing from the closest position of trajectory 'Q' from the starting location of vehicle trajectory 'P' to the closest position of trajectory 'Q' from the end location of vehicle trajectory 'P', e.g., for the three trajectories (A, B, C), considering the closest starting and end positions of trajectory B from trajectories A and C, as shown in Figure 3.

$$\overrightarrow{V_{B-C}} = \langle\, B_9[x] - B_2[x],\ B_9[y] - B_2[y]\,\rangle$$

Here, $B_z[*]$ denotes the image coordinates of the $z^{th}$ position in trajectory B. Similarly, for trajectories A and B,

$$\vec{V}_{B-A} = \langle\, B_{10}[x] - B_3[x],\ B_{10}[y] - B_3[y]\,\rangle$$

The angle difference $A_d$ has been calculated as,

$$A_d(P,Q) = \text{angle between } \vec{V_P} \text{ and } \vec{V}_{Q-P} \text{ where } A_d(P,Q) \in [0,\,360°] \quad (3)$$

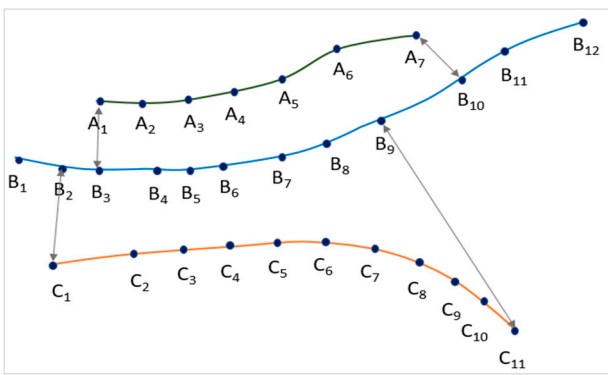

**Figure 3.** Illustration for defining angle similarity.

Since, for trajectories of different patterns and lengths $\vec{V}_{P-Q} \neq \vec{V}_{Q-P}$, hence, the angle similarity $T_s$ is defined as

$$T_s(P,Q) = \begin{cases} A_d(P,Q) \text{ where } L(P) \leq L(Q) \\ A_d(Q,P) \text{ where } L(P) > L(Q) \end{cases} \quad (4)$$

where $L(P)$ is the length of trajectory 'P'. For example, for trajectory 'B' shown in Figure 3,

$$L(B) = \sqrt{\left(B_{12}[x] - B_1[x]\right)^2 + \left(B_{12}[y] - B_1[y]\right)^2}$$

- ■ Proximity of end locations: For vision-based detection systems, depending on their extrinsic calibration, two adjacent movement groups might appear very similar if only the distance and angle similarities are considered. For example, this can be observed in Figure 1a where trajectories in the right-turning lane and the adjacent through movement appear quite similar to each other. However, the right-turn movements travel outside the camera coverage sooner than the through movements, which will lead to different end locations points for right-turn and through movements. Therefore, when different movement streams are densely located within the video frame, using the previous two similarity aspects might lead to misclassification. The end positions of vehicles turning (left or right) or proceeding through an intersection are significantly different from each other (unless inflicted with incomplete trajectory problem) and thereby can serve as an additional factor for improving the classification process. Since this study aims to classify trajectories obtained from vision-based sensors, the end locations of vehicular trajectories were also taken into consideration. In this step, the first rear distance $[D_R]$ is computed.

$$D_R(P,Q) = \sqrt{\left(P_{end}[x] - Q_{end}[x]\right)^2 + \left(P_{end}[y] - Q_{end}[y]\right)^2} \quad (5)$$

Here, $P_{end}[*]$ denotes the coordinates of the end location of trajectory 'P'. The final proximity factor $[F_E]$ is estimated as,

$$F_E(i,j) = \begin{cases} \frac{T_s(P,Q)}{3.6} \times \{D_R(P,Q) - D_S(P,Q)\} & \text{for, } D_R(P,Q) \geq D_S(P,Q) \\ \frac{T_s(P,Q)}{3.6} \times \{D_R(P,Q) - D_S(P,Q)\} & \text{for, } D_R(P,Q) < D_S(P,Q) \text{ and } T_s(P,Q) \leq 15 \\ 0 & \text{for, } D_R(P,Q) \langle D_S(P,Q) \text{ and } T_s(P,Q) \rangle 15 \end{cases} \quad (6)$$

A threshold of 15 degrees is chosen for angular similarity to avoid misclassification in adjacent movement classes. Since both distance and angular similarities are combined in the final proximity, the angular proximity has been scaled down with a factor of 3.6. The final similarity measure $[S(i,j)]$ to be used in the clustering phase is chosen as,

$$S(P,Q) = [\{w_1 \times D_S(P,Q)\} + \{w_2 \times T_s(P,Q)\} + \{w_3 \times F_E(P,Q)\}] \quad (7)$$

A lower value of $S(P,Q)$ therefore indicates higher similarities between the two trajectories. As seen from the above formulations, the final proximity factor will increase $S(P,Q)$ when the trajectories belong to two different movement groups. This increment is proportional to the angle similarity. As a result, for incomplete trajectories which follow the same movement (although, $D_R \geq D_S$), this increment will be negligible.

At traffic intersections where more than one lane is dedicated for moving in a certain direction, multiple vehicular streams are observed under a specific movement type. For these cases, the rear distance $D_R$ can be smaller than the distance similarity $D_S$ within a movement group (Figure 1a). In such scenarios, the proximity factor will decrease the final estimate of $S(P,Q)$ thereby reducing dissimilarities between spatially separated vehicular streams within each movement type.

Linkage Rule: To scale dissimilarities between input trajectories, a linkage criterion is to be chosen for hierarchical clustering. Existing linkage rules include minimum distance or single linkage, maximum distance or complete linkage and unweighted average. The following two considerations governed the choice of linkage criteria at this stage.

At traffic intersections, multiple vehicular streams can be observed for each movement group. The number of vehicle trajectories observed for different movement groups is not always similar. This also applies to different vehicular streams within each movement group. As a result of this, the spread of vehicular streams as obtained from vision-based sensors varies irrespective of their movement direction.

Therefore, the shortest distance or single linkage criterion can be more suitable for this clustering phase. The linkage distance between two clusters $d\left(g_i, g_j\right)$ is obtained as

$$d\left(g_i, g_j\right) = \min(S(P,Q)) \forall P \in g_i, \; Q \in g_j \quad (8)$$

where $g_i$, $g_j$ are two different movement groups obtained from clustering; P and Q are the vehicular trajectories present in those respective movement groups. The movement clusters obtained in this stage are further used to extract modeling trajectories for each movement of interest.

### 3.1.3. Modeling Trajectory Selection

Although classifying vehicular trajectories by clustering is an automated approach, this is not preferable for analyzing a large number of vehicular trajectories (as such cases will create a significant calculation overhead). To overcome this problem, the concept of similarity-based assignment has been used in this study. This concept, which was also utilized in several manual effort-based movement classification methods [32,33], demands the specification of modeling trajectories. Such trajectories represent the general trend of how vehicles move while turning or proceeding through the intersection. Hence, in this stage, one or more trajectories are to be chosen from each identified movement cluster as modeling trajectories.

For an automated selection of modeling trajectories, the identified movement clusters are again grouped into one or more subclusters depending on the number of lanes dedicated for each movement direction. A similar hierarchical clustering process, as discussed above in the "identifying movement clusters" step, is followed for this purpose. However, unlike the previous clustering stage, the objective here is to find intra-cluster lane-specific movement patterns. Therefore, in this clustering phase, the proximity factor is not considered for obtaining the final similarity measure (as it might separate out incomplete trajectories from full-length ones). In addition, vehicle trajectories are not always uniformly spaced within each identified cluster and sometimes even within lane-specific movement groups. As a result, proceeding with the single linkage rule might group trajectories passing through different lanes into one cluster. Since our aim here is to obtain lane-specific trajectory groups, the single linkage criterion is replaced with the average linkage.

<u>Selection criterion</u>—The appropriate selection of modeling trajectories is crucial for the final classification stage. Ideally, the central trajectory should be chosen for this purpose, but since video-based trajectories are curbed with broken and incomplete trajectory problems, the central trajectory found from sub clustering might include incomplete trajectories, which will not serve the intended purpose of modeling trajectory. Therefore, the longest trajectories of the identified subclusters are selected as modeling trajectories for that specific movement of interest.

### 3.1.4. Movement Assignment

In this stage, each incoming vehicle trajectory is compared with the modeling trajectories as identified from the previous stage. The modeling trajectory which is most similar (i.e., resulting in minimum $S(P, Q)$ value) to the input vehicle trajectory is identified and the corresponding movement type is assigned to the incoming vehicle ID.

Similarities of the input trajectory with the modeling trajectories are computed using predefined similarity measure as shown in Equation (7). The proximity factor for end locations is omitted from the final similarity term for the calculation purpose. This is performed for the following reason.

The longest trajectories representing each movement group are used as reference or modeling trajectories. So, the modeling trajectory is not necessarily the central trajectory of each identified movement cluster. Therefore, considering the end-proximity factor might classify the incomplete trajectories of one movement type with an adjacent movement class, the end location of which is more similar to those incomplete trajectories belonging to the first movement type.

Following this similarity-based movement assignment, the movement directions for all vehicle trajectories included in the testing phase are identified, which is further checked with the actual movement directions to evaluate the performance of our proposed algorithm.

### 3.2. Evaluation Metrics

For specifying the evaluation metrics, we first define 4 cases—True Positive (TP), True Negative (TN), False Positive (FP) and False Negative (FN). TP is when an algorithm classifies a movement correctly. TN is when an algorithm rejects certain movement classes correctly. FP is when the algorithm wrongly assigns a movement class and FN is when that algorithm wrongly rejects certain movement classes.

Accuracy is calculated as the ratio of the total number of correct classifications to the total number of trajectories to be classified. It is obtained as

$$\text{Accuracy} = \frac{\text{number of elements correctly classified by model}}{\text{all elements classified by the model}} \tag{9}$$

Balanced accuracy is a performance metric parameter that is used to handle an imbalanced dataset. It is the macro-average of recall scores or True Positive Rate (TPR) per class where

$$TPR = \frac{TP}{TP + FN} \tag{10}$$

The overall balanced accuracy is calculated as the macro-average of the balanced accuracy of all classes.

$$(\text{Balanced Accuracy})_{\text{overall}} = \frac{\sum(\text{Balanced Acuuracy})_i}{\text{Number of classes}} \tag{11}$$

F1 score is another parameter that is considered to acquire performance for those cases where the number of classes is imbalanced. It is the harmonic mean of precision and recall. These can be expressed as,

$$\text{Precision} = \frac{TP}{TP + FP} \tag{12}$$

$$\text{Recall} = \frac{TP}{TP + FN} \tag{13}$$

For the overall F1 score, we take the macro-average for all classes

$$(\text{F1 Score})_{\text{overall}} = \frac{\sum(\text{F1 Score})_i}{\text{Number of classes}} \tag{14}$$

### 3.3. Overview of Methods Used for Comparison

To understand how our proposed algorithm compares with alternative approaches, three existing methods designed for video-based vehicle trajectory classification have been chosen. The first alternative taken into consideration is line-based classification [27]. This is a manual approach that requires the specification of virtual entry–exit lines for each movement of interest. The second one proposed by Liu et al. [32] is a semi-automated approach that looks into the shape similarities of trajectory pairs. This approach follows a rule-based assignment strategy after the manual selection of modeling trajectories. Along with these two methods, an automated approach was also included for comparison purposes. In this method proposed by Hao et al. [38], hierarchical clustering was adopted with the length-scale directive Hausdorff (LSD Hausdorff) distance as a similarity measure. A summary of all methods included in the comparison stage is presented in Table 1.

**Table 1.** Methods Used for Comparison.

| | |
|---|---|
| **Line-based** [27] | ■ Virtual entry–exit line pair manually identified for each MOI.<br>■ Vehicles crossing these line pairs are assigned the corresponding MOI. |
| **Shape-similarity based** [32] | A set of modeling trajectories are manually selected for each MOI from the line-based method.<br>Distance and directional similarities with modeling trajectories are considered.<br>Assigned to the movement where the highest similarity is observed given similarity<br>Values are within predefined limits. |
| **LSD Hausdorff** [38] | ■ Hierarchical clustering with average linkage criteria is used for movement clustering.<br>■ LSD Hausdorff distance is used as a similarity measure. |
| **Proposed Method** | ■ Stop-line identification.<br>■ Hierarchical clustering with single linkage for movement clustering.<br>■ A similarity measure defined in Section 4 was used.<br>■ Sub clustering phase for modeling trajectory selection.<br>■ Similarity-based movement assignment. |

## 4. Data Description

Video data used for the current study were collected from several traffic intersections. The videos were obtained from cameras each facing toward monitoring a single inbound intersection approach. Details regarding the intersection location, video duration and number of vehicle trajectories analyzed are listed in Table 2. This study uses traffic video data obtained from multiple sources. Some of these are openly available, while others are

third-party data. Dubuque intersection data were obtained from the City of Dubuque. The QMUL junction dataset is a publicly available dataset [43]. Among the Indian intersections, the Chandigarh intersection data were obtained from the Chandigarh traffic police. The traffic videos from Hyderabad city are from the openly available 'IITH_Helmet2' dataset [44]. Vehicle movements from the inbound approaches of these intersections were analyzed for evaluating the proposed movement classification algorithm.

**Table 2.** Details Related to Study Intersections.

| Name | Location | No. of Approaches | Video Duration (Hours) | No. of Trajectories |
|------|----------|-------------------|------------------------|---------------------|
| Dubuque | Dubuque, USA | Eight | 4 | 10,000 |
| CHD | Chandigarh, India | One | 1.5 | 600 |
| HYD | Hyderabad, India | One | 1.5 | 400 |
| QMUL | London, UK | One | 1 | 450 |

For this study, we adopted DeepStream SDK [45] to process multiple video streams. We used a ResNet18 model that is pretrained by traffic camera data. It supports the classes of car, bicycle, person, and road signs. We also adopted the tracker based on a discriminative correlation filter (DCF). Our detection-tracking module generates a sequence of bounding box coordinates for each unique vehicle observed in the video stream, thereby forming a trajectory. A visualization of the vehicle trajectory data points obtained from the above-mentioned detection-tracking framework is shown in Figure 4. At each inbound approach, the 1st 30 min video has been used for training purpose. Vehicle trajectories obtained from the remaining video duration have been used for final evaluation. The comparative performance of our proposed algorithm, and the three chosen existing methods on this dataset are presented in the next section.

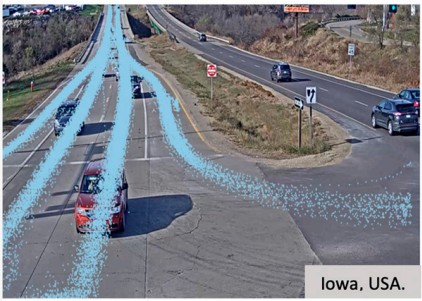 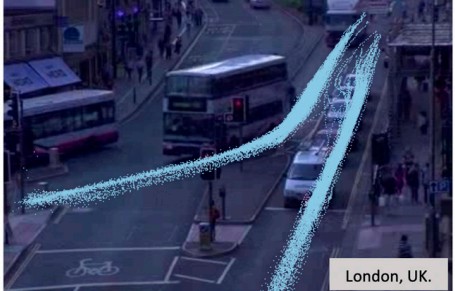
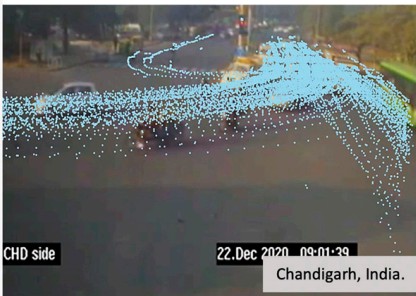 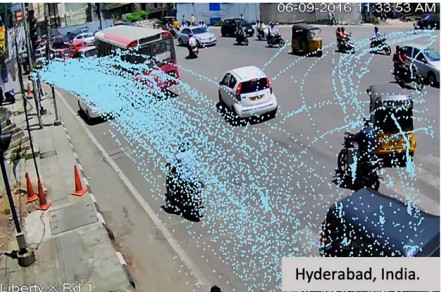

**Figure 4.** Raw Trajectory Data Obtained from different Intersection Approaches.

## 5. Results

As described in the Methodology section, our proposed method identifies the modeling trajectories representative of each movement group following a three-step training phase. These trajectories are then used in the final movement assignment stage. A visual representation of the results obtained at these three steps is shown in Figure 5.

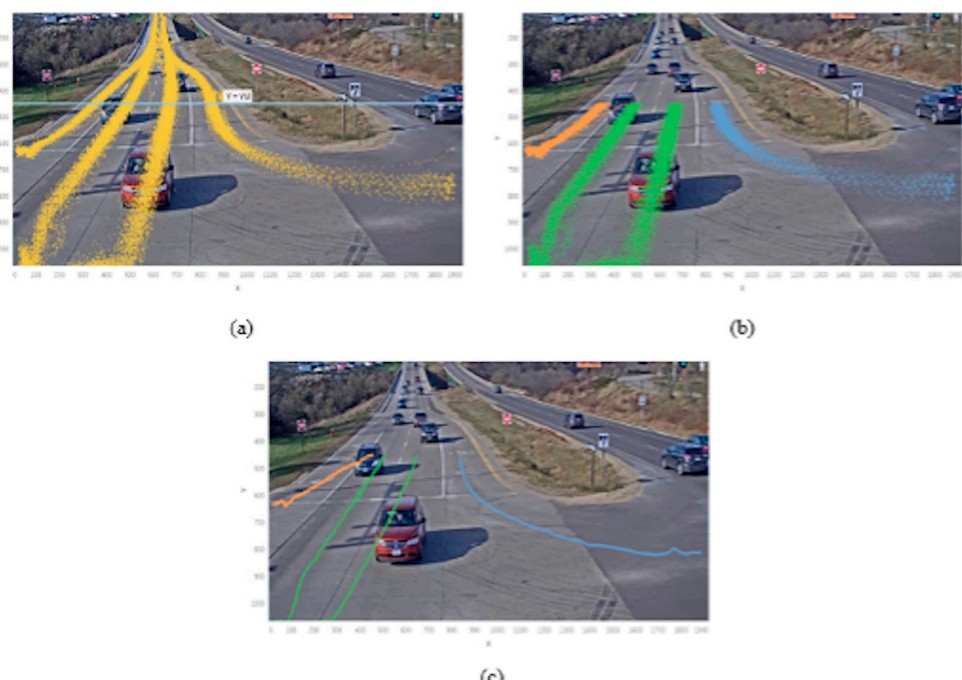

**Figure 5.** Sample results from different stages of the proposed approach: (**a**) stopbar identification, (**b**) movement clusters, (**c**) modeling trajectory.

Figure 5a shows the stopbar location identified from the trajectories of a sample intersection approach, while Figure 5b shows the movement cluster identification from trajectories obtained beyond the stopbar. Figure 5c shows the third step of the proposed approach, modeling trajectory selection. Note here that the raw trajectory dataset (as seen in Figure 5a) used here is extracted from the first 30 min video stream only.

In total, the dataset used in this study contains 11 traffic scenes out of which eight were from two intersections located in Dubuque, Iowa (Table 2). These eight traffic scenes, although from the same location, had different lane configurations. Therefore, to understand how the different classification models (listed in Table 1) perform across similar yet different traffic settings, model performances at Dubuque intersections have been grouped into two categories as NB-SB and EB-WB. The averages of individual model performances in north-bound and south-bound approaches are listed under the first category, while the second category provides the average of individual model performances observed in east-bound and west-bound approaches. Such categories were formed since for north-bound and south-bound approaches, mostly single vehicular streams were observed for each movement group, and these trajectories were sparsely located within the video frame (see Figure 6a,c), whereas in the other category, i.e., in the east and west-bound approaches (see Figure 6b,d), vehicular streams were densely located, and multiple lanes were dedicated to one or more movement groups.

In this study, a clustering-based classification approach is proposed. This proposed approach has been compared with the LSD Hausdorff method, which is the existing state-of-the-art clustering approach that has been used for vehicle movement classification. A comparison of similarity values obtained by the two clustering-based methods is shown in Table 3 in order to assess how the choice of similarity measure in our proposed method helps in the classification process.

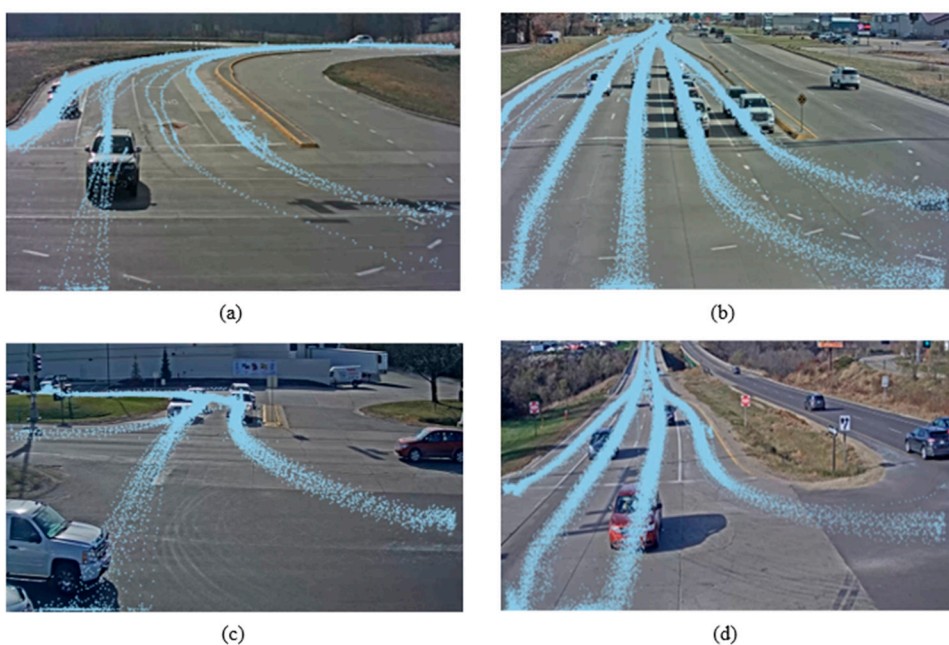

**Figure 6.** Sample Vehicle Trajectories Obtained from Intersections Located in Dubuque: (**a,c**): NB and SB Approaches; (**b,d**) EB and WB Approaches.

**Table 3.** Comparison of similarity measures in the clustering-based classification methods.

| Ground Truth | Similarity Values [1] with respect to Modeling Trajectories of Movement Clusters | | | | | |
|---|---|---|---|---|---|---|
| | Left | Through | Right | Left | Through | Right |
| | LSD Hausdorff | | | Proposed Method | | |
| left_1 | 634 | 706 | **580** | **20** | 758 | 943 |
| left_2 | 854 | 904 | **136** | **44** | 789 | 945 |
| through_1 | 536 | 308 | **214** | 861 | **97** | 561 |
| through_2 | 484 | 193 | **165** | 838 | **115** | 599 |
| right_1 | 754 | 205 | **115** | 900 | 567 | **113** |
| right_2 | 821 | 243 | **127** | 994 | 525 | **41** |

[1] The similarity values are in terms of pixel coordinates. A lower similarity value indicates higher similarity between the trajectories.

The performance of different methods across different intersections as evaluated on the test dataset is shown in Table 4. As seen in Table 4, the proposed method significantly improved the classification performance. In addition, the proposed method could efficiently classify the non-lane-based turning movements as well. A detailed discussion on the performance of these methods across different study locations is included in the next section.

**Table 4.** Comparison of the proposed algorithm with the baseline methods.

| | Balanced Accuracy (%) | | | | | Inference Time [1] |
|---|---|---|---|---|---|---|
| | Dubuque | | HYD | CHD | QMUL | |
| | NB-SB | EB-WB | | | | |
| Line-based | 96.9 | 97.6 | 73.5 | 71.5 | 66.7 | 113 s |
| Shape similarity | 85.5 | 98.3 | 84.5 | 85.9 | 98.2 | 64 s |
| LSD Hausdorff | 89.2 | 69.7 | 96.4 | 93.4 | 81 | 135 s |
| Proposed approach | **99.4** | **99.9** | **98.9** | **95.3** | **100** | **47 s** |

[1] This denotes the computation time required to classify the movements of 1000 vehicle trajectories.

In addition to the vehicle trajectory data, the proposed algorithm utilizes movement-specific lane-level information for the automated selection of modeling trajectories. To understand how our classification method will perform in the absence of such movement-specific lane level information, the proposed classification algorithm has been also assessed with single modeling trajectory selection criteria (i.e., the longest trajectory from each movement group is chosen in this case without any sub clustering. A comparison of the algorithm performance under these two different modeling trajectory selection criteria is shown in Table 5. The balanced accuracy and F1 score values listed here are computed using Equations (9)–(14). It was observed that choosing lane-specific modeling trajectories significantly improved the classification process specially where multiple lanes were dedicated for a single movement direction.

**Table 5.** Effect of Modeling Trajectory selection on Proposed Algorithm Performance.

| Location | Modeling Trajectory | Accuracy (%) | Balanced Accuracy (%) | F1 Score |
|---|---|---|---|---|
| Dubuque NB-SB | Lane-specific | 99.6 | 99.4 | 1.0 |
|  | Single | 99.7 | 99.5 | 1.0 |
| Dubuque EB-WB | Lane-specific | 99.8 | 99.9 | 1.0 |
|  | Single | 88.1 | 95.5 | 0.9 |

## 6. Discussion

The comparative performance of different approaches considered is highlighted below.

The line-crossing-based method did not perform well in Indian intersections where non-lane disciplined behaviors are frequently observed. Along with that, this method could not classify incomplete trajectories that did not cross any pre-specified line pairs and hence led to poor classification accuracies even in structured driving conditions (QMUL).

The shape-similarity-based approach could solve the problem related to incomplete trajectories. However, this method considered fixed distance and angle thresholds. Therefore, vehicle trajectories that follow a different path while moving in a certain direction remain unclassified if they exceed either of the two thresholds. As a result, this method could not adapt to complex Indian driving scenarios.

The third method, LSD Hausdorff, although automated, was only able to classify movement patterns with large spatial separation. As a result, for locations (Dubuque EB-WB, QMUL) where vehicle trajectories were densely located within the video frame, this method was prone to misclassification, and the classification accuracies reduced significantly.

As seen in Table 3, the proposed algorithm performed well across different intersection locations. Among the four different methods used for comparison, this is the only automated approach that could adapt to unstructured driving scenarios and correctly classify non-lane-based traffic maneuvers. The classification process was least influenced by the presence of broken or incomplete trajectories, which is an unavoidable aspect of vision-based trajectories.

If we look at the different model performances in Indian cities as compared to the ones observed in US–UK-based locations, the first two methods (i.e., line-based and shape similarity-based methods) could not achieve high accuracies for Indian locations where a significant amount of non-lane-based driving behavior is observed. The reason behind this is the above two approaches are fixed rule-based classification methods which cannot readily adapt to unstructured traffic environments. In contrast, under such complex traffic scenes, unsupervised or clustering-based methods (LSD Hausdorff, proposed approach) performed significantly better than the previous two rule-based approaches. This is mainly because unlike the previous two approaches, clustering-based methods do not rely on a specific set of predefined rules. Rather, these methods can identify the different movement patterns even when implemented on a new traffic scenario. However, the classification accuracy of these unsupervised methods largely depends on the choice of similarity mea-

sure used in the clustering process. As a result, we could observe a significant difference in the model performances of LSD Hausdorff and the proposed method across different intersection locations. Both of these methods performed well in Indian scenarios, but the similarity measure adopted in the LSD Hausdorff method failed to correctly classify the traffic movements in the US–UK-based locations. In this regard, our considered similarity measure, which is a combination of the distance similarity, angular similarity and end location proximity, worked significantly better compared to the LSD Hausdorff method. The choice of similarity measure in the proposed method could achieve high classification accuracies across all intersection locations analyzed in this study. Although our proposed method performed better than the existing unsupervised classification method, there were a few cases of misclassification as well. At present, for Indian cities, we have analyzed the model performances on very few trajectories. Experimental studies on a larger number of trajectories and several other intersection locations are still required.

From Table 4, it can be observed that for EB–WB locations, where densely located vehicle trajectories have to be classified, selecting multiple modeling trajectories according to the dedicated number of lanes significantly improves the algorithm performance. On the other hand, in NB–SB locations where movement groups are separated by a large spatial margin, the choice of modeling trajectories had a negligible influence over the algorithm performance. Nonetheless, the proposed modeling trajectory selection criteria aided our classification approach to achieve high-accuracy percentages without any scene-specific distance and angle threshold consideration.

There are a few limitations of this study as well. The model performance was evaluated only on four different intersections. Experimental studies on more intersection locations are required to check the robustness of the proposed algorithm and further improvement. The proposed classification framework works on video data obtained from approach-based cameras (i.e., traffic cameras focused on a particular incoming approach). However, opposing traffic movements might also be visible in the camera field of view. At present, we have manually extracted the vehicle trajectories coming through the incoming approach and analyzed the algorithm performance on these trajectories. We intend to explore semantic segmentation-based techniques for the automatic extraction of incoming vehicle trajectories.

## 7. Conclusions

In this study, we tried to address the movement classification task for vehicle trajectories at traffic intersections obtained from vision-based sensors. A fully automated method has been proposed for this purpose considering the inherent shortcomings of video-based detection systems. The new similarity measure defined in the proposed method played an important role for improving the classification performances. Three aspects of similarities were considered for classifying the vision-based movement trajectories. The first consideration was related to the spatial separation or distance similarity. A modified version of the Hausdorff distance was chosen for this purpose, which helped with comparing trajectories of dissimilar length. Next, the angular separation between vehicle tracks was considered. Since the classification task of our interest can involve vehicle tracks of incomplete and dissimilar length even within the same movement group, the angle between adjacent vehicle tracklets was chosen for this. This newly defined angular proximity could efficiently classify vehicle trajectories even when multiple lanes are dedicated for a single movement direction. The end locations of the vehicle trajectories moving in different directions are never similar. Therefore, along with the distance and angular similarity, we had also leveraged the proximity of end locations for defining the similarity measure. The combination of these three similarity aspects carefully designed for non-uniform motion trajectories helped the proposed classification approach perform well in unstructured driving scenarios as well.

Apart from the choice of similarity measure, stopbar identification is a crucial part of our proposed four-step classification framework. Choosing the stopbar helped in filtering

relevant movement trajectories for the subsequent clustering phase. Along with that, vehicle tracks present in the regions both before and after the stopbar were selected in this framework. Such selection criterion was helpful to reduce multi-counting problem occurring due to broken trajectories. Furthermore, the use of lane-specific modeling trajectories helped our classification approach adapt to different traffic scenarios. It has been observed that the appearance of movement patterns within the video frame has a negligible influence on algorithm performance. As a result, the proposed method achieved high accuracies across different intersection locations used for evaluation. Experimental results on the study dataset show that in comparison to existing alternatives, our method has similar or better performance for such movement classification tasks without requiring any manual intervention and thereby can be scalable for implementation at large city-wide or district-wide levels.

The current stopbar identification process is designed for trajectories obtained from a single intersection approach. In the future, we will extend this stopbar identification part for applying in traffic scenes where multiple inbound approaches are visible within a single video frame. In addition, the traffic counts generated from vision-based methods are highly susceptible to detection-tracking accuracy. For this study, ResNet18 and a DCF-based detection-tracking module have been implemented. The tracking output was further utilized to evaluate our classification method. Carrying out more experiments under challenging conditions (i.e., nighttime, heavy occlusion) will help in assessing the impact of tracking on the proposed classification framework.

**Author Contributions:** Conceptualization, U.J. and P.C.; methodology, U.J. and P.C.; formal analysis, U.J., J.P.D.K. and T.H.; investigation, U.J., J.P.D.K., P.C., T.H. and A.S.; resources, P.C. and A.S.; writing—original draft preparation, U.J., J.P.D.K. and T.H.; writing—review and editing, U.J., J.P.D.K., P.C., T.H. and A.S.; supervision, P.C.; funding acquisition, P.C. All authors have read and agreed to the published version of the manuscript.

**Funding:** Our research results are based upon work supported by the Science and Engineering Research Board (SERB) under Sanction Order No. SRB/2020/001708. Any opinions, findings, conclusions, or recommendations expressed in this material are those of the author(s) and do not necessarily reflect the views of the SERB.

**Data Availability Statement:** This study uses traffic video data obtained from multiple sources. Some of these are openly available, while others are third-party data. Dubuque intersection data were obtained from the City of Dubuque and can be provided by the authors with the permission of competent authorities. The QMUL junction dataset is a publicly available dataset [43]. Among the Indian intersections, Chandigarh intersection data were obtained from Chandigarh traffic police and are available from the authors with the permission of Chandigarh traffic police authorities. The traffic videos from Hyderabad city are from the openly available 'IITH_Helmet2' dataset [44].

**Acknowledgments:** The authors would like to thank Dave Ness and Duane Ritcher from City of Dubuque for providing the video data of Dubuque city cameras.

**Conflicts of Interest:** The authors declare no conflict of interest.

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
