# Peer review of "Automated Approach for Computer Vision-Based Vehicle Movement Classification at Traffic Intersections"

_futuretransp, doi:10.3390/futuretransp3020041_

Round 1
Reviewer 1 Report
The author has done in-depth research on vision-based vehicle classification. Compared with the current manual ROI-based operation, it saves manpower and greatly improves the efficiency of automatic classification of traffic intersections. There are several problems:
1. Can the current algorithm distinguish between left and right driving of domestic and foreign vehicles?
2. Please introduce the main defects of radar detection. Can Computer Vision-Based Approaches Avoid It? Have any relevant comparative experiments been carried out?
3. Is there some possibility to combine Radar and Computer Vision?
4. The article mentioned that occlusion will affect the accuracy. Is there any experimental data on the accuracy of the current algorithm on occlusion? In actual traffic scenes, occlusion will be more serious. What accuracy can the algorithm achieve in actual situations?
Reviewer 2 Report
In this paper, the authors proposed an automated classification method for movement-specific classification of vision-based vehicle trajectories. I have some problems, which should be addressed.
1) To verify the effectiveness of the proposed method, some algorithms have been compared, such as Line-based, Shape similarity, and LSD Hausdorff. However, it seems that we do not find the corresponding references. Also, these methods are very old, thus I think that more new methods should be compared.
2) Many datasets are tested. However, their details should be introduced more carefully. Also, whether the download location of the datasets can be provided.
3) There has been a lot of classification methods in the literature, including some deep learning methods. However, the cited works are old, and they were published before 2020. The authors should add more related references, e.g.,
(a)Video moment retrieval with noisy labels. IEEE Transactions on Neural Networks and Learning Systems; (b)Motion Stimulation for Compositional Action Recognition, IEEE Transactions on Circuits Systems and Video Technology;
(c) Recurrent Thrifty Attention Network for Remote Sensing Scene Recognition, IEEE Transactions on Geoscience and Remote Sensing;
(d) Multi-view Learning with Robust Double-sided Twin SVM with Applications to Image Recognition; IEEE Transactions on Cybernetics
(e) Expansion-squeeze-excitation fusion network for elderly activity recognition, IEEE TCSVT
Reviewer 3 Report
(1) In subsection 3.1, it is suggested to add a description of how vehicle tracks are coded.
(2)In subsection 3.1.2, the symbol definition of the trajectory is suggested to be unified as a, b, c or P, Q, M or p, q, m.
(3) In Section 4, please add an experimental scheme to verify the similarity measure in hierarchical clustering, such as drawing the linkage process diagram of hierarchical clustering.
(4) In Section 4, please add the execution time comparison experiment of various clustering methods to verify the real-time performance of the algorithm, such as Hausdorff, DTMD, etc.
Reviewer 4 Report
Quality of the method may depend on the behavior of drivers, style of driving, organisation of roads and crossroads. Differences between US and India have been already uderlined. Lack of deeper discussion about influence of different model intersections (different countries) on the quality of algorithm. From my point of view, this is a report from the research done. Partial, but not exhaustive.
Reviewer 5 Report
The topic of the article Automated Approach for Computer Vision-based Vehicle Movement Classification at Traffic Intersections is relevant and interesting. The authors have a good idea, but the elaboration of the article is not very good.
Details to be corrected:
1) The purpose of the article must be formulated in the introduction.
2) Formula 1 is not clear and incorrect, because both max and minimum are written.
3) Formula 2 is also incorrect because the explanation (where...) does not match the formula itself.
4) In lines 399-411, the formulas are presented out of order and not according to the requirements.
5) Tables 1-4 should be corrected and submitted as required. Line numbers appearing in Tables 2 and 3.
6) It is suggested to expand the discussion section. It is important for the authors to identify any study limitations.
7) References are incorrect and not in accordance with the requirements. More literature sources from prestigious journals such as Symmetry, Sustainability and others need to be added.
8) Some literature sources are too old, e.g., at positions 3, 27, 32.
9) At the end of the article, I missed further research on this topic.
Round 2
Reviewer 2 Report
I suggest accepting this paper since may problems have been solved.
Author Response
Thank you for dedicating your valuable time to providing feedback on our manuscript. We really appreciate your insightful comments on our paper.
Reviewer 5 Report
The authors have revised almost everything in the article based on my comments and suggestions. There are still some inaccuracies left.
Please look at lines 539-542. Two tables are presented here - Table 4 and Table 5. Comments are required after them, because the chapter cannot end with tables without any explanation.
Sonclusions section should also be expanded. Limitation should be moved to the discussion section.
